# Effect of Meropenem on Conjugative Plasmid Transfer in *Klebsiella pneumoniae*

**DOI:** 10.3390/ijms252313193

**Published:** 2024-12-08

**Authors:** Daria A. Kondratieva, Julia R. Savelieva, Maria V. Golikova

**Affiliations:** Department of Pharmacokinetics & Pharmacodynamics, Gause Institute of New Antibiotics, 11 Bolshaya Pirogovskaya Street, 119021 Moscow, Russia; goawaymrway@gmail.com (D.A.K.); savmos80@mail.ru (J.R.S.)

**Keywords:** antibiotic-induced conjugal transfer of plasmids, *Enterobacterales*, Pseudomonas aeruginosa, meropenem, carbapenemases, hollow-fiber infection model

## Abstract

Plasmid-mediated resistance is a major mechanism that contributes to the gradual decline in the effectiveness of antibiotics from different classes, including carbapenems. Antibiotics can significantly contribute to the efficiency of plasmid transfer between bacterial strains. To investigate the potential effect of an antibiotic on the efficacy of conjugative plasmid transfer, we conducted mating experiments with *Klebsiella pneumoniae* strains. Donor strains of *K. pneumoniae* that carry plasmids with *bla*_KPC_ or *bla*_OXA-48_ carbapenemase genes and recipient plasmid-free *K. pneumoniae* strains were used in matings. Matings were conducted on the agar with or without meropenem at 1/8×, 1/4×, or 1/2×MIC against the respective recipients. In the second part of our study, we investigated the pharmacodynamic properties of meropenem against transconjugant strains of *K. pneumoniae*, which were obtained in the first part of this study. As a result, at a concentration equivalent to 1/8×MIC, meropenem primarily inhibited conjugation among *K. pneumoniae* strains, while at a concentration equal to 1/2×MIC, it facilitated conjugation. Transconjugants derived from *K. pneumoniae* with intermediate MICs failed to respond to simulated treatment with meropenem using prolonged infusion and a high-dose regimen. This finding suggests that such transconjugants may potentially pose a risk if involved in an infectious process.

## 1. Introduction

Antibiotic resistance in bacteria is an increasing concern, especially due to the emergence of strains that possess plasmids containing antibiotic resistance genes. Plasmid-mediated resistance is a major mechanism that contributes to the gradual decline in the effectiveness of antibiotics from different classes, including carbapenems, which are the first-line agents used in the management of critically ill patients [1]. Unfortunately, over the past decade, resistance to carbapenems in Gram-negative bacteria has emerged as a significant challenge, particularly among carbapenem-producing *Klebsiella pneumoniae* strains [2,3,4]. These organisms can easily spread among hospitalized patients and pass carbapenem resistance genes located on plasmids to other Gram-negative bacteria through horizontal gene transfer. Two of the most commonly encountered carbapenem resistance genes are *bla*_KPC_ and *bla*_OXA-48_ [5,6,7].

The phenomenon of horizontal gene transfer through conjugation was first described almost a century ago [8], but there is still a great deal to be learned regarding what factors influence the efficacy of this process and how it can be controlled. Antibiotics can significantly contribute to the efficiency of plasmid transfer between bacterial strains. Several studies have been conducted in this area [9,10,11,12,13,14]. The researchers in these studies have shown that the antibiotic not only acts as a selection factor for cells that contain plasmids, increasing their number, but also has the potential to induce the uptake of plasmids by the recipient through intricate cellular mechanisms linked to the upregulation of genes involved in the SOS response [9,14]. However, there are also studies that challenge the notion that antibiotics can promote conjugation [11,12]. Therefore, despite numerous studies on this topic [9,10,13,14], there is still ambiguity regarding the precise role of antibiotics in this process. Therefore, we have decided to conduct research in order to collect additional data on the potential effect of an antibiotic on the efficacy of conjugative plasmid transfer. In order to achieve this, we used three donor strains of *K. pneumoniae* that carry plasmids with *bla*_KPC_ or *bla*_OXA-48_ carbapenemase genes, in addition to two recipient meropenem-susceptible plasmid-free *K. pneumoniae* strains.

In the second part of our study, we investigated the pharmacodynamic properties of meropenem against transconjugant carbapenemase-producing strains of *K. pneumoniae*, which were obtained in the first part of this study. To this end, we simulated therapeutic concentrations of meropenem achieved in the epithelial lining fluid after a high-dose, prolonged infusion regimen [15], using the HFIM [16]. This simulation is clinically significant as nosocomial pneumonia caused by carbapenem-resistant *K. pneumoniae*, which produces carbapenemases, may occur in hospitalized patients who especially require effective antibiotic treatment [17,18]. To assess the potential effectiveness of meropenem in such cases, it is essential to assume meropenem pharmacokinetics at the site of infection (epithelial lining fluid as the site of pneumonia infection), as they may differ substantially from plasma levels. We have chosen meropenem as the subject of our research, as it is a front-line antibiotic within the carbapenem class commonly used in the treatment of critically ill patients [19]. This choice underscores the relevance and potential importance of our study.

This study aims to explore the dual effect of meropenem on plasmid transfer efficiency and its potential to suppress the growth of *K. pneumoniae* transconjugants under simulated clinical exposure in an in vitro hollow-fiber infection model.

## 2. Results

### 2.1. Mating Experiments

To investigate whether antibiotics may influence conjugation frequency, we carried out a series of mating experiments using three pairs of *K. pneumoniae* strains. Plasmid donor *K. pneumoniae* strains carried broad-host-range conjugative plasmids with carbapenemase genes. We have decided to concentrate our research on the investigation of how plasmids with carbapenemase genes are transferred, as resistance conferred by these genes negatively affects the effectiveness of carbapenems. These criteria were met by three strains, which are producers of KPC (*K. pneumoniae* 565) and OXA-48 (*K. pneumoniae* 485 and *K. pneumoniae* 38) carbapenemases. These are currently some of the most prevalent carbapenemases globally. The recipient strains, in turn, were susceptible to carbapenems and did not carry plasmids containing carbapenemase genes: collection *K. pneumoniae* ATCC 700,603 and clinical *K. pneumoniae* 188 strains. Therefore, we investigated six donor–recipient pairs, respectively: (1) *K. pneumoniae* 565–*K. pneumoniae* ATCC 700603; (2) *K. pneumoniae* 565–*K. pneumoniae* 188; (3) *K. pneumoniae* 485–*K. pneumoniae* ATCC 700603; (4) *K. pneumoniae* 485–*K. pneumoniae* 188; (5) *K. pneumoniae* 38–*K. pneumoniae* ATCC 700603; and (6) *K. pneumoniae* 38–*K. pneumoniae* 188. Matings were conducted on the Luria Agar with or without meropenem. Based on the conventional approach, the sub-inhibitory antibiotic concentrations in the agar were adjusted to correspond to 1/8, 1/4, or 1/2 of the MIC for the respective recipient used in the mating experiments in order to avoid inhibiting its growth. The results of these experiments are presented in Figure 1. In general, conjugation frequencies varied widely (from 10⁻⁷ to 10⁻^2^ transconjugants per recipient) for different bacterial donor–recipient combinations and different plasmids, and this was dependent on the concentration of meropenem in the agar. For the pair *K. pneumoniae* 565–*K. pneumoniae* ATCC 700603, an increase in plasmid transfer was observed as the concentration of meropenem in the agar increased: the higher the concentration of meropenem, the higher the conjugation frequency. To ensure the observed differences are statistically significant, we used a *p*-value of 0.05. Assuming the calculated *p*-values, a statistically significant increase in plasmid transfer was observed between the following cases: control–1/8×MIC, 1/4×MIC–1/2×MIC, and control–1/2×MIC. In other donor–recipient combinations, the dynamics of conjugation frequency were more complex and depended on the concentration of meropenem. When the concentration was 1/8×MIC, conjugation frequency decreased sharply in four donor–recipient pairs (*K. pneumoniae* 565–*K. pneumoniae* 188, *K. pneumoniae* 485–*K. pneumoniae* ATCC 700603, *K. pneumoniae* 485–*K. pneumoniae* 188, and *K. pneumoniae* 38–*K. pneumoniae* ATCC700603), compared to in the control group. Subsequently, with an increase in meropenem concentration, the frequency of plasmid transfer increased gradually until reaching a maximum at 1/2×MIC. The *p*-values for all cases indicated that the observed differences were statistically significant. The dynamics of conjugation frequency in donor–recipient pair *K. pneumoniae* 38–*K. pneumoniae* 188 was slightly different from that of other donor–recipient pairs in terms of its dependence on meropenem concentration. An increase in meropenem concentration up to 1/4×MIC in agar did not lead to any significant change in the frequency of plasmid transfer. However, when the meropenem concentration reached 1/2×MIC, there was a sharp increase in the frequency of transfer compared to both the control, 1/8×MIC, and 1/4×MIC. In all cases, the conjugation frequency at 1/2×MIC was always higher than in the control group without meropenem, and this difference was statistically significant.

As illustrated in Figure 1, a relatively more significant increase in conjugation efficiency was observed when the initial plasmid transfer was relatively lower. To visually analyze this phenomenon, a graph has been constructed, which is presented in Figure 2.

The graph compares the frequency of conjugation in the absence of meropenem (control group) with the ratio of the conjugation frequency at 1/2 × MIC of meropenem to its control value (CF_1/2×MIC_/CF_CONT_). This ratio indicates how many times the frequency of conjugation has increased in the presence of meropenem. As can be seen from the graph, in cases where the *K. pneumoniae* donor strains produced OXA-48 carbapenemases (strains 485 and 38), there was an inverse consistence between the initial conjugation frequency in the control group and the enhanced conjugation rate in the presence of 1/2×MIC of meropenem.

It is worth noting that all transconjugants obtained from the matings were tested for their ability to continue growing in the presence and the absence of meropenem. Their plasmid carriage was confirmed by PCR, and their meropenem susceptibility was assessed. In all instances, the transconjugant strains showed sustained growth either in the presence or absence of the antibiotic and carried the plasmids (Appendix A, represents PCR data from experiments with the recipient strain *K. pneumoniae* ATCC 700603 as an example). The MICs of meropenem for transconjugant strains generated through mating experiments are presented in Table 1 below.

When comparing the conjugation frequency between donor–recipient pairs, we observed that the donor strain *K. pneumoniae* 565 exhibited relatively high conjugation rates compared to those of other donor strains. This was particularly evident in the *K. pneumoniae* 565–*K. pneumoniae* ATCC 700603 combination, which demonstrated higher rates of plasmid transfer, at least an order of magnitude greater than those of other pairs. Based on these findings, we decided to further investigate the conjugation of the *K. pneumoniae* 565 strain with bacteria from other species, including *Escherichia coli* and *Pseudomonas aeruginosa*. The results of these mating experiments are presented in Figure 3.

As shown in the figure, in mating experiments with *K. pneumoniae* and *E. coli*, a similar trend to that observed with the *K. pneumoniae* 565–*K. pneumoniae* ATCC 700,603 pair was seen: the higher the meropenem concentration in the agar, the higher the conjugation frequency. However, there was one exception where conjugal plasmid transfer did not differ significantly at meropenem concentrations of 1/8×MIC and 1/4×MIC (*p* = 0.06). In contrast, in mating with *K. pneumoniae* and *P. aeruginosa*, an opposite trend was seen, with increasing meropenem concentration leading to a gradual decrease in conjugation frequencies. However, these differences were not as pronounced.

While analyzing the ability of *E. coli* and *P. aeruginosa* transconjugants for further growth on agar containing meropenem, we found that *E. coli* transconjugants mostly exhibited weak growth. Only a few examples were capable of growing in the presence of the antibiotic, suggesting the presence of a fitness cost. In contrast, *P. aeruginosa* transconjugants grew well. PCR analysis revealed plasmid loss in both *E. coli* and *P. aeruginosa* transconjugant variants (in approximately 25–100% of the analyzed samples), which was unexpected, as *P. aeruginosa* has previously been shown to exhibit good growth in the presence of meropenem while predominantly losing plasmids. MICs of meropenem against *E. coli* ATCC 25922 and *P. aeruginosa* ATCC 9027 plasmid-containing transconjugants that showed good growth on agar with or without of meropenem were 2 µg/mL and 8 µg/mL, respectively.

### 2.2. Pharmacodynamic Experiments

In the second part of our study, we sought to evaluate the viability of *K. pneumoniae* transconjugants obtained in mating experiments under exposure of meropenem using a hollow-fiber infection model. The main component of the hollow-fiber infection model is a bioreactor that replicates an infection site. Within this bioreactor, bacterial cells are cultivated and the human pharmacokinetics of antibiotics are simulated. This is accomplished through the controlled infusion of an antibiotic into the model, followed by the dilution of the antibiotic-containing medium with fresh medium, so that the pharmacokinetic profile of the antibiotic in the bioreactor mimics that of humans. As a result, antibiotic concentrations fluctuate constantly, initially increasing (during drug administration) and then decreasing (following drug elimination). This allows for the study of bacterial response to antibiotics under conditions that closely resemble clinical ones. Bacterial cells may be killed, proliferate, or develop resistance to the antibiotic.

We considered it to be essential to assess the capacity of the transconjugant strains to survive during simulated treatment and their ability to maintain carbapenemase production to combat the antibiotic. To this end, therapeutic levels of meropenem in epithelial lining fluid were simulated (lung infection model). This allowed us to evaluate the efficacy of meropenem at the site of infection, where the drug’s concentration is approximately half that of plasma. In these experiments, two *K. pneumoniae* ATCC 700603 transconjugant strains, which were characterized by the highest and lowest meropenem MICs of 4 µg/mL and 0.5 µg/mL, respectively, were used. These strains were derived from matings with donor strains *K. pneumoniae* 565 and *K. pneumoniae* 38 and are referred to as *K. pneumoniae* tc1 and *K. pneumoniae* tc2, respectively. In these experiments, we conducted a simulation of the highest dose regimen of meropenem with the aim of evaluating its potential effectiveness against *K. pneumoniae* strains that produce KPC and OXA-48 carbapenemases. Assuming transconjugants were cultured in the HFIM until 10^8^ CFU/mL density before meropenem was infused, we ensured they maintained carbapenem resistance and plasmids by susceptibility testing and PCR, respectively. For this purpose, samples from the HFIM were obtained before the antibiotic was infused. Additionally, experiments were conducted with the recipient strain *K. pneumoniae* ATCC 700603, which served as a control. As illustrated in Figure 4, despite exposure to meropenem, there was a significant increase in the bacterial count and the maximum level was reached by the 24th hour for strain *K. pneumoniae* tc1 with the meropenem MIC of 4 µg/mL. This rapid growth of the total bacterial population was accompanied by an intense selection of meropenem-resistant cells across all resistance levels, ranging from 2× to 16×MIC. Conversely, strain *K. pneumoniae* tc2 with the meropenem MIC of 0.5 µg/mL was completely eliminated from the system after 48 h of exposure to meropenem. Recipient strain *K. pneumoniae* ATCC 700603 was eliminated from the system in the first 6 h (Appendix A).

## 3. Discussion

It is generally believed that antibiotics may facilitate horizontal gene transfer [9,10,13,14]. However, it is important to note that the claim that the use of antibiotics promotes conjugation is not universally accepted. There is some conflicting evidence regarding whether antibiotics can actually promote plasmid transfer. The underlying mechanism that is thought to be involved in antibiotic action on plasmid transfer involves activating the excision of transferable genes from the host chromosome, triggering typical SOS gene expression, which in turn leads to increased stress conditions that can act as inducers, increasing mutagenesis and horizontal gene transfer [9,14]. However, there are several studies that suggest that the antibiotic’s effect is only selective [11,12]. Despite the mentioned contradictions, we believe that the potential of antibiotics to induce conjugation highlights the significance of further research in this area. This may help us to develop ways to manipulate these processes in order to minimize the risks of the horizontal transfer of antibiotic-resistant genes among bacteria. In an effort to gather additional data on the topic, we conducted a series of experiments aimed at exploring whether the presence of an antibiotic in the medium may induce the transfer of conjugative plasmids between donor and recipient bacterial strains of *K. pneumoniae*. To this end, meropenem was added to the agar plates on which plasmid donors and recipients were paired. Sub-inhibitory concentrations of the antibiotic were selected in order to ensure that recipient susceptibility to meropenem would not inhibit its growth. Such inhibition has been demonstrated in the work of Yang Lu et al., published in 2017 [13], where conjugation between *E. coli* strains at gentamicin levels equal to the MIC significantly decreased compared to that of the control. In fact, the levels of antibiotics within the human body during treatment may be below the inhibitory levels. For example, meropenem may be used in patients undergoing abdominal surgery, in which the doses and duration of administration may be relatively low [20]. In such cases, the concentration of meropenem can decrease significantly before the next dose, reaching sub-inhibitory levels by the end of the dosing interval.

As can be seen in Figure 1, the presence of an antibiotic in the medium in all cases had an effect on conjugation efficiency, compared to that of the control. The influence was dependent on both the strains used for mating and the meropenem concentration. In all instances where the meropenem concentration was highest (at half the MIC), there was a clear increase in plasmid transfer efficiency compared to that of the control group. Therefore, we can conclude that the antibiotic clearly promoted this process. A similar trend was observed in the interaction between *K. pneumoniae* and *E. coli*. According to published data, the plasmid transfer in *E. coli* increased when the bacteria were grown in the presence of 1/2×MIC of cefotaxime or amoxicillin [21]. The use of sub-inhibitory doses of florfenicol or oxytetracycline also significantly enhanced conjugative plasmid transfer in *E. coli* strains [10]. There are several other similar examples of this phenomenon [22,23,24,25]. However, we also observed that when antibiotic levels were reduced to 1/8× or 1/4×MIC, plasmid acquisition in most strains was less frequent than in the control group, leading to a decrease in conjugation frequency. This phenomenon may be explained by the fact that the meropenem concentration of 1/8× and 1/4×MIC already has a negative impact on the growth of recipient cells, which in turn reduces their ability to participate in conjugation. However, this antibiotic effect does not seem to be enough to trigger the typical SOS response mechanism in recipient cells, which, according to current data, is responsible for antibiotic-induced conjugation [9,14]. However, there was an exception for the *K. pneumoniae* 565–*K. pneumoniae* ATCC 700603 donor–recipient pair, where meropenem treatment resulted in increased plasmid transfer in all cases. It is worth noting that an inverse consistency was observed between the conjugation rate in the control group (without meropenem) and its increase in the presence of meropenem at 1/2×MIC: the lower the conjugation without the antibiotic, the more significant the increase in conjugation in the presence of the antibiotic compared to that of the control. This is illustrated in Figure 2. However, these initial findings require further validation, as this trend has only been seen in a limited number of donor–recipient combinations, which is not sufficient for a definitive conclusion.

In experiments involving *K. pneumoniae* in combination with *P. aeruginosa*, we observed an unexpected result with meropenem. Specifically, the higher the concentration of meropenem, the lower the frequency of conjugation. Based on the analysis of three pairs of strains, where the donor was *K. pneumoniae* 565 and the recipients were different species of bacteria, we found similar trends for *K. pneumoniae* and *E. coli* from the family *Enterobacterales*. However, this trend was not observed for *P. aeruginosa*. In the context of *P. aeruginosa* when it is conjugated with *K. pneumoniae*, the impact of antibiotic treatment on plasmid transfer appears to be more complex and therefore warrants further investigation, which we plan to conduct in our future research. A potential explanation for these unexpected findings is that *P. aeruginosa* might experience plasmid loss, and as a result, it may employ additional resistance mechanisms, in addition to plasmid conjugation, in the presence of the meropenem. Actually, *P. aeruginosa* is characterized by a flexible genome and the capacity to implement various mechanisms of antibiotic resistance, except for carbapenemase production. These include efflux pumps and modifications in the expression and/or structure of porins [26,27,28]. It is essential to extend our research to encompass a greater number of donor–recipient combinations. Moreover, a more comprehensive analysis is required.

In the pharmacodynamic part of our research, we evaluated the effectiveness of meropenem against two carbapenemase-producing *K. pneumoniae* transconjugant strains with high and low MICs of meropenem. In these experiments, we simulated the maximum clinical dose regimen of meropenem to assess both its antibacterial activity against carbapenem-resistant bacteria and the ability of these bacteria to resist drug exposure. This is an important consideration because bacteria that acquire plasmids through conjugation can subsequently lose these plasmids [29,30,31]. The reasons for this phenomenon can be attributed to various factors, including the individual properties of plasmids, their acquisition costs, and/or their fitness costs in recipient cells [32,33,34]. As was obtained, the *K. pneumoniae* transconjugants exhibited varying abilities to survive under the meropenem exposure. As would be expected, the strain with the lowest meropenem MIC of 0.5 µg/mL was eliminated from the system, as it did not exhibit the ability to resist meropenem. Conversely, the *K. pneumoniae* transconjugant with the meropenem MIC of 4 µg/mL showed intensive growth and the development of resistance to meropenem. We have previously conducted a study on the pharmacokinetics of meropenem in relation to transconjugant strains generated through mating experiments involving *K. pneumoniae* and either *K. pneumoniae* or *E. coli* [35]. In that study, we modeled lower concentrations of meropenem in peritoneal fluid achieved with a dosing regimen consisting of 500 mg administered in a 0.5 h infusion. For this specific investigation, however, we employed a different dosing strategy using an extended infusion of meropenem (3 h) and a maximum dose of 2000 mg to assess the efficacy of this high-dose meropenem regimen against transconjugant strains that produce carbapenemases. Unfortunately, a higher dose of meropenem did not achieve the desired effect on the *K. pneumoniae* strain tc1 with an MIC of 4 µg/mL. This suggests that this strain has a high capacity to produce carbapenemase enzymes. Consequently, there is a potential for these initially highly susceptible organisms that do not produce carbapenemases to acquire plasmids carrying carbapenemase genes, which may make them resistant to meropenem treatment.

Our study has several limitations. Firstly, it did not include a wide variety of bacterial strains with varying MICs or other classes of carbapenemases, such as metallo-β-lactamases. Secondly, a more comprehensive investigation using genetic techniques could also assist in answering some of the questions that arose during the course of this study.

## 4. Materials and Methods

### 4.1. Antimicrobial Agents, Bacterial Strains, and Susceptibility Testing

Meropenem powder was purchased from Sigma-Aldrich (St. Louis, MO, USA). Three clinical isolates of *K. pneumoniae* were used as plasmid donors in mating experiments: 38 and 485 (meropenem MICs of 16 and 32 µg/mL, respectively, isolated in Moscow in 2011 year and in Saint Petersburg in 2012, respectively [36]) and 565 (meropenem MIC of 64 µg/mL, isolated in Saint Petersburg in 2011 [37]). Strains *K. pneumoniae* 38 (ST147) and 485 (ST395) carry plasmids with *bla*_OXA-48_ carbapenemase genes (pOXAAPSS2 and pOXAAPSS1, respectively; both IncL; KU159086.1 and KU159085.1, respectively). Strain *K. pneumoniae* 565 (ST273) carries plasmid with *bla*_KPC-2_ carbapenemase genes (pKPCAPSS plasmid, IncFII, KP008371). The plasmid pOXAAPSS1 also carries genes encoding resistance to penicillins and early cephalosporins (*bla*_TEM-1_). The plasmid pKPCAPSS also carries resistance genes to penicillins and early cephalosporins (*bla*_TEM-1_), macrolides (*mphA*, *mrx*, *mphR*), and fluoroquinolones (*qnrS1*). Strains of *E. coli* ATCC 25922, *K. pneumoniae* ATCC 700603, *K. pneumoniae* 188 (clinical isolate), and *P. aeruginosa* ATCC 9027 were used as recipients in mating experiments. Meropenem MICs for all recipient strains with except for *P. aeruginosa* (meropenem MIC of 0.25 µg/mL) were equal to 0.03 µg/mL. Before and after each testing or experiment, carbapenemase production was verified for each bacterial strain using a modified carbapenem inactivation method [38].

Susceptibility testing was carried out using broth microdilution techniques with a standard inoculum of approximately 5 × 10^5^ CFU/mL. Meropenem MICs were determined according to standard recommendations using cation-supplemented Mueller–Hinton broth (CSMHB) (Becton Dickinson, Franklin Lakes, NJ, USA) [39]. Before reading, microplates were incubated at 37 °C for 18–20 h. MIC values in each case were obtained at least in triplicate, and modal MICs were estimated.

### 4.2. Mating Experiments

The protocol of mating experiments followed to obtain transconjugants is summarized in Figure 5.

To distinguish between donor, recipient, and transconjugant cells, all the recipient strains (*E. coli* ATCC 25922, *K. pneumoniae* ATCC 700603, *K. pneumoniae* 188, and *P. aeruginosa* ATCC 9027) were incubated on the media with rifampicin to produce rifampicin-resistant variants (MIC > 512 µg/mL). For all experiments, these rifampicin-resistant mutants were used.

Bacteria were grown in Luria Broth (LB, Becton Dickinson, Franklin Lakes, NJ, USA) and Luria Agar (LA, Becton Dickinson, Franklin Lakes, NJ, USA) media [40] at 37 °C. Matings were performed overnight on the LA plates with or without meropenem (concentrations of 1/8×, 1/4×, and 1/2 times the MIC). Briefly, the 1:1 mixture of the donor and recipient in the late logarithmic growth phase was plated on the LA surface and incubated at 37 °C for 18–20 h. The mixed growth was then scraped from the plate surface and then resuspended in 1 mL of saline, and to quantify the numbers of donor, recipient, and transconjugant cells, the cell mixture was diluted as appropriate. Then, 100 µL samples were spread on appropriate selective plates with meropenem and/or rifampicin at the following final concentrations: meropenem (equal to 4×MIC of recipient) and rifampicin (200 µg/mL). Parent strains were plated in parallel with the matings and then processed similarly to the matings as controls. Isolated colonies from matings presumed as transconjugant and parental strains from controls were used to identify recombinants and parental forms. The conjugation efficiency was assessed by the conjugation frequency: the ratio of the number of CFU of transconjugants per mL to the number of CFU of recipients per mL.

Possible transconjugants were screened by streaking colonies from the selection plate with meropenem and rifampicin onto a fresh plate with the same antibiotics to look for growth. The plasmid acquisition by recipient strains was confirmed by PCR with primers specific to genes encoding plasmid replication proteins and relaxases (positions 53829–54896 and 30249–32228 in KU159085.1; positions 1039–2159 and 22713–23548 in KU159086.1; 77542–78366 and 83825–84614 in KP008371.1). PCR was performed according to a standard protocol for amplification of fragments with a size of 1 kb [41].

### 4.3. In Vitro Dynamic Model and Operational Procedure Used in the Pharmacodynamic Experiments

The hollow-fiber infection model was used to evaluate meropenem pharmacodynamics and to conduct growth control experiments. The operational procedure is described in detail elsewhere [42]. Briefly, the model consisted of three connected flasks, the first containing fresh CSMHB, the second (central) compartment with CSMHB and the antibiotic, and the third (peripheral) compartment with a hollow-fiber bioreactor (Fresenius dialyzer, model AV400S, Fresenius Medical Care AG, 328 Bad Homburg, Germany) containing bacterial culture plus the antibiotic that represents the infection site. Peristaltic pumps (Masterflex, Cole-Parmer Instrument Co., Chicago, IL, USA) were used to continually replace the antibiotic-containing medium with fresh medium and to circulate the medium between and within the compartments. The schematic illustration of the HFIM is provided in Appendix A. The system was filled with sterile CSMHB and placed in an incubator at 37 °C. The inoculum of an 18 h culture of *K. pneumoniae* (transconjugant or recipient (control)) was injected into the hollow-fiber bioreactor to produce a bacterial concentration of 10^8^ CFU/mL. After a 2 h incubation, samples were obtained to determine the starting bacterial concentration. Antibiotic dosing and sampling were processed automatically, using computer-assisted controls. Then, the meropenem-containing medium was pumped to the central camera as a 3 h infusion to ensure the desired time courses of meropenem as it occurs in human epithelial lining fluid. In parallel, a continuous flow of meropenem-containing medium was pumped from the central compartment to the peripheral, and the flow returned to the central compartment. The concentration of meropenem in both compartments remained the same. The peripheral compartment of the hollow-fiber infection model was sampled for bacterial concentrations over 120 h immediately after the end of infusion (3 h) and at the 6th hour of the dosing interval.

In each experiment, the bacteria-containing medium from the central unit of the model was sampled to determine bacterial concentrations throughout the observation period. Samples (100 µL) were serially diluted as appropriate, and 100 µL was plated onto Mueller–Hinton agar plates, which were placed in an incubator at 37 °C for 24 h. The lower limit of accurate detection was 1 × 10^2^ CFU/mL (equivalent to 10 colonies per plate).

To monitor the time courses of antibiotic-resistant sub-populations of *K. pneumoniae* in the pharmacodynamic experiments, the central unit of the model was multiply-sampled throughout the observation period (120 h). The samples were serially diluted, if necessary; plated on Mueller–Hinton agar (MHA) with 2×, 4×, 8×MIC, and 16×MIC of meropenem; and incubated for 24–48 h at 37 °C. The viable counts were screened visually for growth. The lower limit of detection was 10 CFU/mL (equivalent to at least one colony per plate).

### 4.4. Antibiotic Dosing Regimens and Simulated Pharmacokinetic Profiles

Meropenem treatment mimicked the therapeutic dosing regimen: 2 g administered every 8 h, as a 3 h intravenous infusion. A mono-exponential profile in epithelial lining fluid (ELF) after thrice-daily dosing of meropenem with a half-life (t1/2) of 1.4 h was simulated [15] for 5 consecutive days with a total of 15 infusions. The pharmacokinetic parameter values were as follows: CMAX = 32.4 µg/mL, 24 h area under the concentration–time curve (AUC) = 375 (mg × h)/L. Before all pharmacodynamic simulations, the system was calibrated and preliminary in vitro pharmacokinetic experiments in CSMHB without bacteria were conducted.

### 4.5. Statistical Analysis

The reported MIC data were obtained by calculation of the respective modal values. The resultant conjugation frequency was calculated as arithmetic mean ± standard deviations for three replicate experiments. The data from each group were analyzed for statistically significant differences (*p* < 0.05) in the data mean values between the groups using SigmaPlot 12 statistical software (Systat Software Inc., headquartered in San Jose, CA, USA) by paired two-sample *t*-test.

In pharmacodynamic and growth control experiments, bacterial count data were calculated as arithmetic mean ± standard deviations for three replicate experiments. Based on these data, kinetic growth and time-kill curves were constructed. Assuming that the coefficient of variation of the log CFU/mL data was less than 10%, in order to facilitate figure viewing, we decided not to include data point error bars in order to avoid interference with the kinetic curves.

## 5. Conclusions

The current study has identified several key observations. Firstly, there appears to be a trend in the varying effects of sub-inhibitory concentrations of meropenem on conjugation efficiency among *K. pneumoniae* strains, ranging from inhibition to promotion. Specifically, at a concentration equivalent to 1/8 of the MIC, meropenem primarily inhibited conjugation among *K. pneumoniae* strains, while at a concentration equal to 1/2 of the MIC, it facilitated conjugation. Similar facilitation was observed for pairs of *K. pneumoniae* and *E. coli*, but not for pairs involving *K. pneumoniae* and *P. aeruginosa*. Secondly, transconjugants derived from *K. pneumoniae* with intermediate MICs failed to respond to simulated treatment with meropenem using prolonged infusion and a high-dose regimen. This finding suggests that such transconjugants could potentially pose a risk if involved in an infectious process.

## Figures and Tables

**Figure 1 ijms-25-13193-f001:**
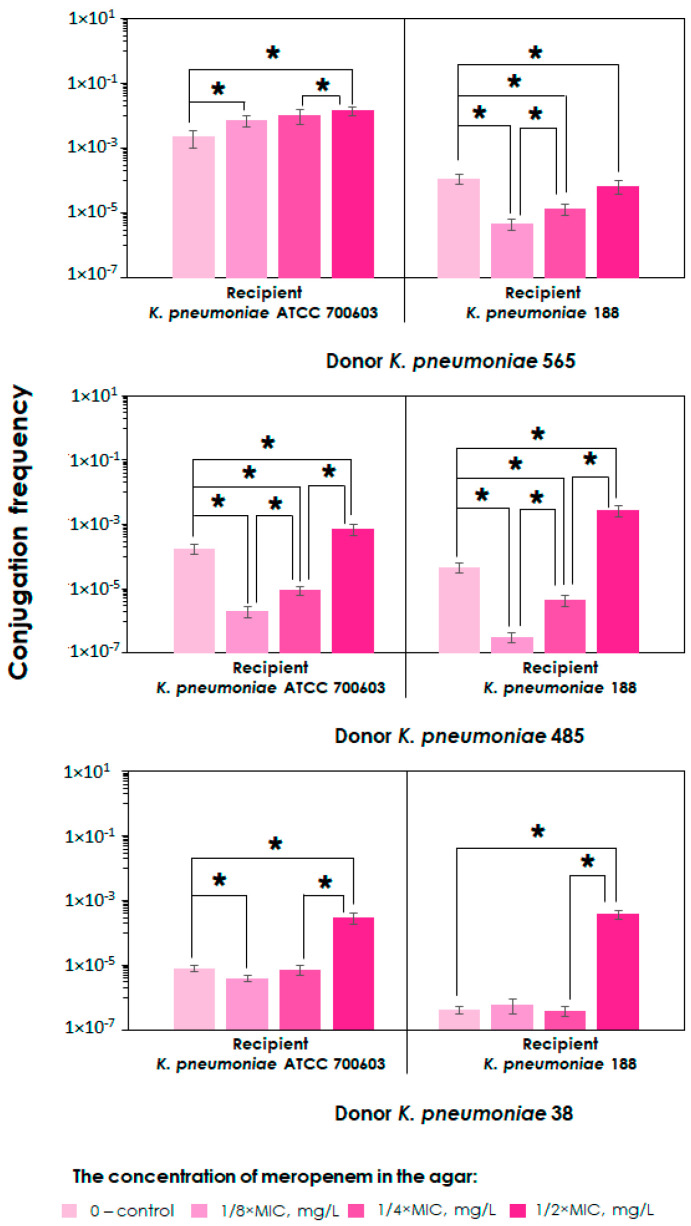
Conjugation frequencies in *K. pneumoniae–K. pneumoniae* donor–recipient pairs with or without (control group) meropenem. Asterisk (*) represents statistically significant differences (*p* < 0.05). Error bars represent standard deviations (n = 3).

**Figure 2 ijms-25-13193-f002:**
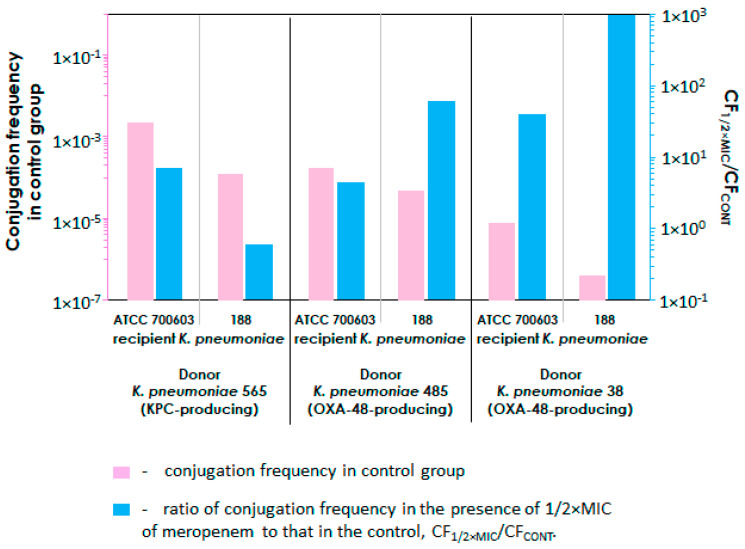
Conjugation frequency without meropenem and its increase under the meropenem exposure as reflected by the ratio CF_1/2×MIC_/CF_CONT_ (ratio of conjugation frequency in the presence of 1/2×MIC of meropenem to conjugation frequency in the control).

**Figure 3 ijms-25-13193-f003:**
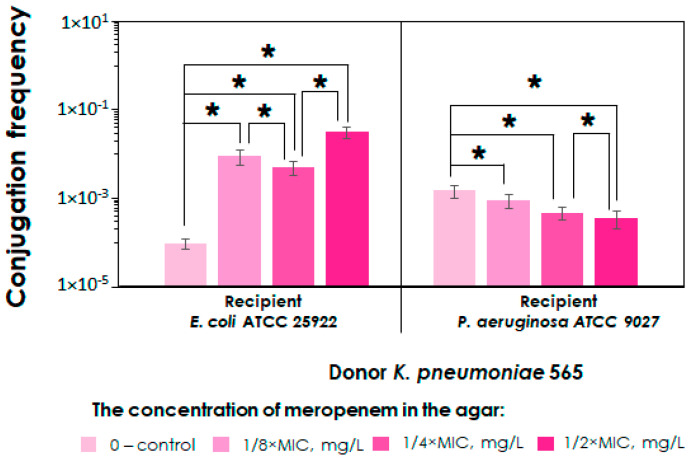
Conjugation frequencies in *K. pneumoniae* 565–*E. coli* ATCC 25922 and *K. pneumoniae* 565– *P. aeruginosa* ATCC 9027 donor–recipient pairs with or without (control group) meropenem. Asterisk (*) represents statistically significant differences (*p* < 0.05). Error bars represent standard deviations (n = 3).

**Figure 4 ijms-25-13193-f004:**
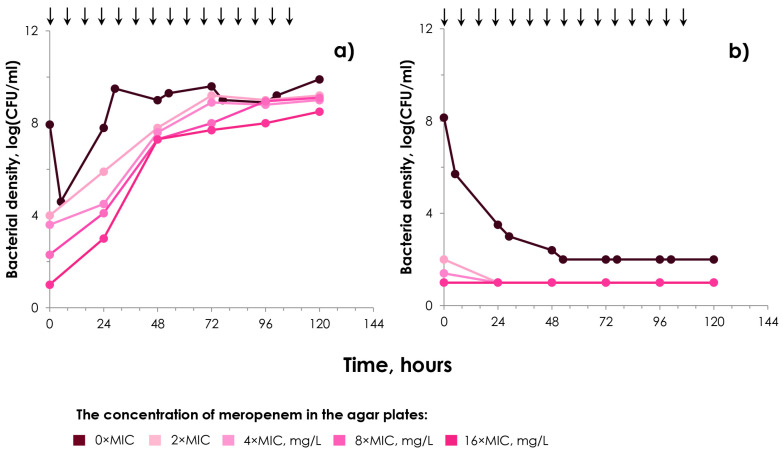
Time courses of the total bacterial population (0 × MIC) and meropenem-resistant (2×, 4×, 8× and 16 × MIC) sub-populations of transconjugant carbapenemase-producing strains of *K. pneumoniae* in pharmacodynamic experiments (high-dose meropenem exposure): (**a**)—experiments with transconjugant strain tc1 with meropenem MIC of 4 µg/mL; (**b**)—experiments with transconjugant strain tc2 with meropenem MIC of 0.5 µg/mL. Data are presented as arithmetic means (n = 3). Arrows indicate the start of meropenem infusion.

**Figure 5 ijms-25-13193-f005:**
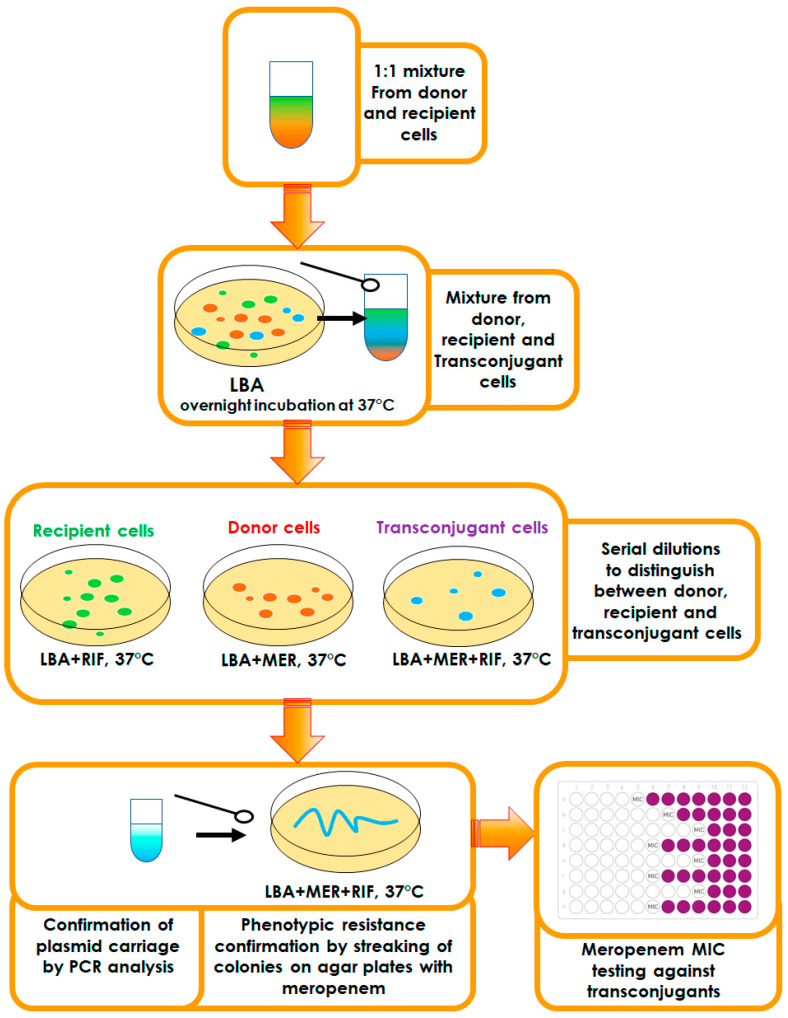
Schematic representation of the conjugation protocol.

**Table 1 ijms-25-13193-t001:** Meropenem MICs for transconjugant strains obtained in mating experiments (in µg/mL).

Donor–Recipient Pairs		Recipient *K. pneumoniae* Strain	
		ATCC 700603	188
Donor *K. pneumoniae* strain	565	4	2
	485	0.5	0.5
	38	0.5	0.25

## Data Availability

Data contained within the article.

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
