# Peer review of "Effect of Meropenem on Conjugative Plasmid Transfer in Klebsiella pneumoniae"

_ijms, 2024, doi:10.3390/ijms252313193_

Round 1
Reviewer 1 Report
Comments and Suggestions for Authors
In the manuscript by D. A. Kondratieva and colleagues, the authors studied the effect of the antibiotic meropenem on conjugative plasmid transfer in Klebsiella pneumoniae. They conducted mating experiments with donor and recipient plasmid-free strains with and without meropenem at various concentrations. They also investigated the pharmacodynamic properties of meropenem against transconjugant strains of K. pneumoniae.
The strength of this work lies in its exploration of how different concentrations of meropenem may promote or inhibit conjugative plasmid transfer between different pairs of mating strains. The manuscript is overall easy to follow; however, the writing, data presentation, and interpretation could be improved for better readability and impact, as outlined below:
Major comments:
1. In section 2.1, the reviewer suggests the authors specify the choice of three donor and two recipient strains at the beginning before jumping into the results or Figure 1.
2. Line 83, the reviewer asks the authors to provide reasoning for why the concentration was not increased to the MIC like the reference [4].
3. Line 87, the reviewer asks the authors to clarify whether the strain mentioned as K. pneumoniae 38-188 is a typo and should be 385-188.
4. The observed reduction in conjugation frequency at 1/8 MIC is intriguing. The authors should propose possible explanations or mechanisms for this phenomenon.
5. Line 110, the reviewer suggests replacing vague terminology like “trend towards a relationship” with a clear description of the observed relationship.
6. Line 111, the reviewer suggests specifying which strains are OXA-48 carbapenemase-producing and elaborate on the mention of KPC carbapenemase genes in the abstract for consistency.
7. Line 115, the reviewer suggests including the PCR results as the supplementary figure.
8. In section 2.2, the reviewer suggests providing more context for focusing on epithelial lining fluid and lung infection models, particularly their relevance to carbapenem resistance in K. pneumoniae.
9. In section 2.2, the reviewer suggests explaining why the hollow-fiber infection model was chosen and how it simulates in vivo conditions for antibiotics like meropenem.
10. Figure 1 and 3, missing y-axis label.
11. Figure 4, add clear labels to the colored curves and indicate if replicates were performed.
12. Lin 184-187, consider discussing other potential factors, such as plasmid type and environmental conditions, that might influence the observed horizontal gene transfer.
13. Line 194-195, Under what circumstances might these sub-inhibitory levels occur in real-world clinical or environmental scenarios?
14. The reviewer saw some studies that have shown that antibiotics may not directly regulate the efficiency of horizontal gene transfer. Instead, they influence the overall dynamics of conjugation by altering the competitive landscape among bacterial populations. Have the authors considered studying competitive landscape dynamics and mixed communities?
15. The reviewer wonders if the authors investigated whether plasmids retained in recipient strains when meropenem is removed.
Minor comments:
1. Line 51, “K. pneumoniae” needs to be italicized. Please check throughout the manuscript.
2. Line 56-60 writing is a bit confusing. The reviewer suggests rewriting it in a way similar to “This study investigates the dual role of meropenem in influencing plasmid transfer efficiency and its ability to suppress the growth of transconjugants under simulated clinical conditions”
3. Line 154, “it essential”. Missing “is” in between.
Reviewer 2 Report
Comments and Suggestions for Authors
This manuscript describes investigations of effect of meropenem on plasmid transfer from carbapenemase producing Klebsiella pneumoniae. Topic of this manuscript is an interesting issue, however, some parts of this study are unclear.
Comments
1) In this study three carbapenemase producing Klebsiella pneumoniae strains were used. These strains could be better described:
Which other resistance determinants were present in these strains appart from carbapenemase gene?
Which clones do these three K. pneumoniae strains belong? (Sequence type should be mentioned).
2) In this study conjugation assay was performed with 1:1 ratio of donor and recipient strains. What was the reason to apply only this ratio. In the scientific literature there are different protocolls available that apply 1:2, 1:3 or even 1:4 ratio of donor and recipient strains.
3) Were there any addtional resistance phenotype detected during conjugation assay in transconjugant strains? I mean, were there any additional resistance gene transfered to transconjugant strain the indicate co-transfer and co-localisation on a plasmid beside a carbapenemease gene?
4) Please, check it in all over the manuscript that bacterial names are written in italic form (Klebsiella pneumoniae). At the first appearance in the abstract and in the main text full name (Klebsiella pneumoniae), but later on in the text short form (K. pneumoniae) must be written.
Reviewer 3 Report
Comments and Suggestions for Authors
Dear authors and editor,
The authors of the manuscript addressed an important and exciting topic. Understanding the phenomenon of resistance related to the transfer of plasmids responsible for antibiotic resistance is important.
Though the harnessing of antibiotics is one of the most significant human innovations, their efficacy is continuously eroded by the craftiness of their microbial targets. Once a single bacterium mutates to become resistant to antibiotics, it can transfer that resistance to other bacteria around it through a process known as horizontal gene transfer. One of the primary vehicles for gene transfer among bacteria is small circular pieces of DNA or plasmids. Plasmids can be transferred through direct physical contact between bacteria in a process known as conjugation, which helps bacteria share their antibiotic-resistant genes with their neighbors.
Although conjugation is well-understood on a molecular level, how it plays out in the environments that bacteria inhabit, rather than the lab, is much less clear. One exceptionally versatile pathogen, Klebsiella pneumoniae, is particularly interesting for studies on resistance gene sharing because it forms so-called persister reservoirs in its hosts.
Can antibiotics significantly contribute to the efficiency of plasmid transfer between bacterial strains? Please elaborate on this argument so that the readers understand.
Line 43-44: Please insert the references.
Why meropenem and not another antibiotic?
The introductory part contains much information related to the study's purpose and objectives and some data about the importance of plasmids responsible for resistance to antibiotics in the case of Klebsiella pneumoniae bacteria.

Round 2
Reviewer 1 Report
Comments and Suggestions for Authors
The reviewer appreciates the author's efforts in addressing the comment. The updated manuscript showed great improvements in the data interpretation and presentation.
My last small comment is to see whether you can find some references for line 61-69.
Reviewer 3 Report
Comments and Suggestions for Authors
Dear authors,
Thank you for the answers. I appreciate the authors' effort to respond punctually to the recommendations and comments from the primary revision process. Through the significant revision, the quality of the manuscript has improved considerably. The description is more precise, the results are rendered explicitly, and other essential clarifications are highlighted. Considering the above, from my point of view, the manuscript in its current state can be considered for publication.
Best regards, Reviewer
